# Quantitative Assessment of Interfacial Interactions Governing Ultrafiltration Membrane Fouling by the Mixture of Silica Nanoparticles (SiO_2_ NPs) and Natural Organic Matter (NOM): Effects of Solution Chemistry

**DOI:** 10.3390/membranes13040449

**Published:** 2023-04-21

**Authors:** Yuqi Sun, Runze Zhang, Chunyi Sun, Zhipeng Liu, Jian Zhang, Shuang Liang, Xia Wang

**Affiliations:** 1Shandong Provincial Key Laboratory of Water Pollution Control and Resource Reuse, Shandong Key Laboratory of Environmental Processes and Health, School of Environmental Science and Engineering, Shandong University, Qingdao 266237, China; yuqisun0821@163.com (Y.S.); xiaotur953904076@gmail.com (R.Z.); scy3735@163.com (C.S.); 202132853@mail.sdu.edu.cn (Z.L.); zhangjian00@sdu.edu.cn (J.Z.); 2College of Safety and Environmental Engineering, Shandong University of Science and Technology, Qingdao 266000, China; 3State Key Laboratory of Microbial Technology, Shandong University, Qingdao 266237, China; ghwx@sdu.edu.cn

**Keywords:** ultrafiltration, membrane fouling, silica nanoparticles (SiO_2_ NPs), natural organic matter (NOM), interfacial interaction, xDLVO theory

## Abstract

Mixtures of silica nanoparticles (SiO_2_ NPs) and natural organic matter (NOM) are ubiquitous in natural aquatic environments and pose risks to organisms. Ultrafiltration (UF) membranes can effectively remove SiO_2_ NP–NOM mixtures. However, the corresponding membrane fouling mechanisms, particularly under different solution conditions, have not yet been studied. In this work, the effect of solution chemistry on polyethersulfone (PES) UF membrane fouling caused by a SiO_2_ NP–NOM mixture was investigated at different pH levels, ionic strengths, and calcium concentrations. The corresponding membrane fouling mechanisms, i.e., Lifshitz–van der Waals (LW), electrostatic (EL), and acid–base (AB) interactions, were quantitatively evaluated using the extended Derjaguin–Landau–Verwey–Overbeek (xDLVO) theory. It was found that the extent of membrane fouling increased with decreasing pH, increasing ionic strength, and increasing calcium concentration. The attractive AB interaction between the clean/fouled membrane and foulant was the major fouling mechanism in both the initial adhesion and later cohesion stages, while the attractive LW and repulsive EL interactions were less important. The change of fouling potential with solution chemistry was negatively correlated with the calculated interaction energy, indicating that the UF membrane fouling behavior under different solution conditions can be effectively explained and predicted using the xDLVO theory.

## 1. Introduction

Silica nanoparticles (SiO_2_ NPs) have been widely applied in numerous fields (e.g., food industry, drug delivery, and material science), due to their high surface-area-to-volume ratio and good biocompatibility [1,2,3]. As a result, there are increasing amounts of SiO_2_ NPs eventually entering the natural aquatic environment through multiple pathways. Previous studies have indicated that SiO_2_ NPs can damage the cell structures of organisms and influence their activities [4,5,6]. Moreover, natural organic matter (NOM), abundant in surface water, can interact with SiO_2_ NPs and form SiO_2_ NP–NOM mixtures [7]. NOM is a complex pool of organic compounds, mainly including humic substances, proteins, and polysaccharides [8,9]. It can influence the surface properties and stability of SiO_2_ NPs and subsequently increase the health risks to both ecosystems and humans [10]. Therefore, it is important to effectively remove SiO_2_ NP–NOM mixtures from natural water.

Ultrafiltration (UF) has been proven to be effective for removing SiO_2_ NPs and NOM [11,12,13]. For example, Springer et al. [13] reported that the retention rate of SiO_2_ NPs by UF membranes was greater than 99.6%. However, membrane fouling of UF membranes, increasing the operational and maintenance cost, remains the principal obstacle constraining their widespread application [14,15]. Tian et al. [14] investigated the combined effect of NOM and SiO_2_ NPs on UF membrane fouling. They reported that the fouling resistance caused by SiO_2_ NP–NOM mixtures was much higher than that caused by the individual foulants (i.e., SiO_2_ NPs or NOM) alone. Similarly, Martin et al. [15] also observed that there was an obvious synergistic effect between SiO_2_ NPs and NOM on UF membrane fouling. These studies demonstrated the significant potential of SiO_2_ NP–NOM mixtures to foul UF membranes, making a full understanding of this fouling behavior desirable. Moreover, methods for mitigating UF membrane fouling have also been put forward, such as the control of solution chemistry having important effects on fouling behavior [16]. Nevertheless, in the case of UF membrane fouling by SiO_2_ NP–NOM mixtures, the solution chemistry has not been investigated.

The significant effect of solution chemistry on membrane fouling by SiO_2_ NP–NOM mixtures has been reported in nanofiltration (NF) [17] and forward osmosis (FO) systems [18]. For example, Arkhangelsky et al. [18] found that FO membrane fouling by SiO_2_ NP–NOM mixtures was aggravated in the presence of calcium ions. In the case of UF, though the fouling effect of solution chemistry remains unknown in terms of SiO_2_ NP–NOM mixtures, its importance in UF membrane fouling has been observed to some extent, in terms of NOM or SiO_2_ NPs alone [19,20]. Dong et al. [19] carried out UF of NOM solutions at various pH levels and found that membrane fouling decreased significantly with increasing pH from 5.5 to 7.5. Singh et al. [20] reported that feed water with low pH and high ionic strength induced the greatest UF membrane fouling by SiO_2_ NPs. It is therefore reasonably expected that solution chemistry will have an important effect on UF membrane fouling by SiO_2_ NP–NOM mixtures. Since SiO_2_ NP–NOM mixtures exhibit a synergistic fouling effect and significantly more serious UF membrane fouling behavior compared with NOM or SiO_2_ NPs alone [14,15], their membrane fouling responses to changes of solution chemistry will be considerably different from those of NOM or SiO_2_ NPs alone and thus deserve further investigation.

The mechanisms governing UF membrane fouling by SiO_2_ NP–NOM mixtures have not yet been seriously investigated. To date, these mixtures have been shown to contribute generally to membrane pore blocking and cake formation, which are commonly occurring mechanisms in all membrane fouling processes [14,15]. Essentially, membrane fouling results from the adhesion of suspended foulants to the clean membrane surface in the initial filtration stage, followed by the cohesion of suspended foulants to the foulants already attached to the membrane surface. The tendency of adhesion and cohesion has been found to be determined by three major interfacial interactions (i.e., Lifshitz–van der Waals (LW), acid–base (AB), and electrostatic (EL) interactions), which can be quantitatively evaluated using the extended Derjaguin–Landau–Verwey–Overbeek (xDLVO) theory [21,22]. Although it has been successfully used to explain membrane fouling mechanisms in many other UF processes [23,24], the xDLVO theory has not yet been applied to elucidate the interfacial interactions governing UF membrane fouling by SiO_2_ NP–NOM mixtures, not to mention under scenarios of different solution chemistry conditions. The lack of such useful information greatly hinders the development of more precise control strategies for UF membrane fouling by SiO_2_ NP–NOM mixtures.

The main objective of this study was to systematically investigate the effect of solution chemistry, including pH, ionic strength, and calcium concentration, on UF membrane fouling caused by a SiO_2_ NP–NOM mixture. The xDLVO theory was, for the first time, adopted to elucidate the corresponding membrane fouling mechanisms by quantitatively evaluating the interfacial interactions (i.e., LW, AB, and EL interactions) between the SiO_2_ NP–NOM mixture and UF membrane, and within the SiO_2_ NP–NOM mixture. The acquired results contribute to a more complete understanding of UF membrane fouling caused by SiO_2_ NP–NOM mixtures, and, particularly, provide valuable quantitative insights into the underlying membrane fouling mechanisms.

## 2. Materials and Methods

### 2.1. UF Membrane

A commercial flat sheet UF membrane (Shanghaimosu Filtering Equipment Co. Ltd., Shanghai, China) made of polyethersulfone (PES) was adopted in this study. According to the manufacturer, it has a molecular weight cut-off (MWCO) of 100 kDa and an effective surface area of 50.26 cm^2^. Prior to fouling experiments, the PES membrane was soaked in deionized (DI) water for 24 h with several intermediate water changes for sufficient removal of impurities or additives. A fresh membrane was used for each run.

### 2.2. Preparation of Model Foulants

The SiO_2_ NPs (Aladdin, Shanghai, China) employed in this study had a purity of 99.5% and a primary particle size of 15 nm. A stock solution (1 g/L) was prepared by dissolving powdery SiO_2_ NPs into DI water, and then sonicating for 15 min using a sonication bath. Commercially available humic acid (HA), bovine serum albumin (BSA), and sodium alginate (SA) (Sigma-Aldrich, St. Louis, MO, USA) were used as the representative humic substances, proteins, and polysaccharides in the NOM, respectively. All organic foulants were received in powder form. Stock solutions (1 g/L) were prepared by dissolving the HA, BSA, and SA in DI water and then stirring for 24 h, followed by filtering through a 0.45 μm cellulose acetate membrane to remove insoluble matters. The stock solutions were stored in a refrigerator at 4 °C.

To simulate the NOM composition in natural water, the concentrations of HA, BSA, and SA were set to be 8.0, 2.0, and 2.0 mg/L, respectively [25,26]. In the SiO_2_ NP–NOM solution, the concentrations of SiO_2_ NPs and NOM were 1.5 and 12 mg/L, respectively. For pH effect evaluation, the solution pH was adjusted to 3, 5, 7, and 9 by adding 0.1 mol/L HCl and 0.1 mol/L NaOH as needed, while the ionic strength was maintained constant at 10 mM. Four different NaCl concentrations of 10, 30, 60, and 100 mM were used to investigate the effect of ionic strength at a solution pH of 7. SiO_2_ NP–NOM solutions with different calcium concentrations of 0, 0.3, 0.6, and 1.0 mM were prepared to study the effect of divalent cations at a constant pH of 7 and ionic strength of 10 mM.

### 2.3. Membrane Fouling Experiments

A dead-end stirred cell (MSC300, Mosu Corp., Shanghai, China) with an inner diameter of 8 cm and an effective filtration area of 50.26 cm^2^ was employed to study UF membrane fouling by the SiO_2_ NP–NOM mixture under different solution conditions. The schematic diagram of the experimental setup can be found in our previous work [27]. To avoid concentration polarization and automatic settling of foulants, the stirring speed was set at 180 rpm throughout the experiments. Membrane fouling experiments were performed at a room temperature of 20 ± 1.0 °C, and the transmembrane pressure, provided by a nitrogen gas cylinder, was set at a constant 100 kPa. Prior to each membrane fouling experiment, deionized water was filtered through the UF membrane for 20 min to stabilize the membrane and obtain the clean water flux *J*_0_. Then, the feed solution was driven through the membrane, and permeate flux was continuously monitored using a top-loading electronic balance (BL-1200S, Setra Systems, Everett, WA, USA) connected to a computer. 

To better understand the membrane fouling behaviors under different solution conditions, the entire filtration period was separated into initial and later stages, corresponding to the adhesion and cohesion stages, respectively. The separation of the two stages was carried out according to the methods described by Sun et al. [27]. Fouling potential (*K*) was adopted as the indicator for membrane fouling evaluation. It is defined as the reduction in normalized flux caused by a unit mass of SiO_2_ NP–NOM mixture filtered through the membrane [23]. The fouling potential in the initial (adhesion) and later (cohesion) stages was determined according to the following equation.
(1)$$K=\frac{\Delta (J/J_{0})}{\Delta {V\times C}_{0}}$$
where *K* is the fouling potential (mg^−1^), Δ*J/J*_0_ is the difference in the normalized flux between the beginning and end of the filtration stage (%), Δ*V* is the difference in the effluent volume between the beginning and end of the filtration stage (L), and *C*_0_ is the concentration of the SiO_2_ NP–NOM solution (mg/L).

### 2.4. Analytical Methods

The contact angles of the PES UF membrane and the SiO_2_ NP–NOM mixture were measured using a contact angle analyzer (JC2000C Contact Angle Meter, Shanghai Zhongchen Experiment Equipment Co. Ltd., Shanghai, China) based on the sessile drop method [23,28,29]. The SiO_2_ NP–NOM mixture was first deposited on the PES UF membrane using the dead-end stirred-cell filtration system, and then subjected to contact angle measurement [30]. The measurement of contact angle was performed at the time of two seconds after the liquid drop formed on the clean PES UF membranes or the PES UF membranes covered with SiO_2_ NP–NOM mixture. Three probe liquids of known surface tension values (ultrapure water, glycerol, and diiodomethane) were selected for contact angle measurements. To ensure the reliability of results, at least seven contact angles at different locations were determined and averaged.

The zeta potentials of the PES UF membrane and the SiO_2_ NP–NOM mixture were determined using a zeta potential analyzer (SurPASS, Anton Paar, Graz, Austria) and a zetasizer (3000HSa, Malvern Instruments, Malvern, UK), respectively. Each data value is the average of five measurements. All measurements were performed at 20 °C. The particle size of the SiO_2_ NP–NOM mixture under different solution conditions was measured via dynamic light scattering measurement (DLS; BI-200SM/BI-9000, Brookhaven, Holtsville, NY, USA).

### 2.5. The xDLVO Theory

According to the xDLVO theory [31], the total interaction energy between a planar membrane and a spherical colloidal particle is accounted as the sum of the Lifshitz–van der Waals (LW), Lewis acid–base (AB), and electrostatic double-layer (EL) interactions, which can be written as follows:(2)$$U_{\mathrm{mlf}}^{\mathrm{TOT}}{=U}_{\mathrm{mlf}}^{\mathrm{LW}}{+U}_{\mathrm{mlf}}^{\mathrm{AB}}{+U}_{\mathrm{mlf}}^{\mathrm{EL}}$$
where $U_{mlf}^{TOT}$ is the total interaction energy between the membrane surface and the foulants; $U_{mlf}^{LW}$, $U_{mlf}^{AB}$, and $U_{mlf}^{EL}$ represent the LW, AB, and EL components of the interaction energies, respectively. The subscripts *m*, *l*, and *f* correspond to the membrane, the liquid, and the foulants, respectively.

#### 2.5.1. Surface Tension Parameters

To determine the interaction energy, surface tension parameters ($\gamma_{s}^{\mathrm{LW}}$, $\gamma_{s}^{+}$, and $\gamma_{s}^{-}$) of the PES UF membrane and the SiO_2_ NP–NOM mixture were calculated using the extended Young equation [21,32,33], which requires contact angle measurements to be performed using three probe liquids with known surface tension parameters and can be written as follows:(3)$${(1+\cos\theta )\gamma }_{l}^{\mathrm{TOT}}=2(\sqrt{\gamma_{s}^{\mathrm{LW}}\gamma_{l}^{\mathrm{LW}}}+\sqrt{\gamma_{s}^{+}\gamma_{l}^{-}}+\sqrt{\gamma_{s}^{-}\gamma_{l}^{+}})$$
where the subscripts *s* and *l* represent the solid surface (i.e., membrane and foulants) and the probe liquid, respectively. *θ* is the contact angle, and *γ^LW^*, *γ^+^*, and *γ*^−^ are the LW component, electron-acceptor component, and electron-donor component, respectively. *γ^TOT^* is the total surface tension, which can be expressed as follows:(4)$$\gamma^{\mathrm{TOT}}{=\gamma }^{\mathrm{LW}}{+\gamma }^{\mathrm{AB}}$$
(5)$$\gamma^{\mathrm{AB}}=2\sqrt{\gamma^{+}\gamma^{-}}\;,$$
where *γ^AB^* is the acid–base surface tension component.

To avoid the effect of the surface roughness on the contact angle, the measured contact angle (*θ_m_*) was corrected using the Wenzel equation:(6)$$\cos\;\theta =\frac{{\cos\;\theta }_{m}}{r}=\frac{{\cos\theta }_{m}}{1+\mathrm{SAD}}$$
where *r* is a correction factor for the contact angle and *SAD* is the surface area difference.

#### 2.5.2. Adhesion Free Energy and Cohesion Free Energy

The adhesion free energy signifies the interaction energy per unit area between two surfaces when the separation of them is close to the minimum equilibrium cut-off distance. Moreover, the LW and AB adhesion energy per unit can be evaluated via the surface tension parameters calculated above and expressed as follows:(7)$${\Delta G}_{d_{0}}^{\mathrm{LW}}=2(\sqrt{\gamma_{l}^{\mathrm{LW}}}-\sqrt{\gamma_{m}^{\mathrm{LW}}})(\sqrt{\gamma_{f}^{\mathrm{LW}}}-\sqrt{\gamma_{l}^{\mathrm{LW}}})$$
(8)$${\Delta G}_{d_{0}}^{\mathrm{AB}}=2\sqrt{\gamma_{l}^{+}}(\sqrt{\gamma_{m}^{-}}+\sqrt{\gamma_{f}^{-}}-\sqrt{\gamma_{l}^{-}})+2\sqrt{\gamma_{l}^{-}}(\sqrt{\gamma_{m}^{+}}+\sqrt{\gamma_{f}^{+}}-\sqrt{\gamma_{l}^{+}})-2(\sqrt{\gamma_{m}^{+}\gamma_{f}^{-}}+\sqrt{\gamma_{m}^{-}\gamma_{f}^{+})}$$
where the subscript *d*_0_ is the minimum equilibrium cut-off distance (0.158 ± 0.009 nm). If two surfaces are composed of identical membranes or foulants, the sums of Δ$G_{d}^{LW}$0and Δ$G_{d}^{AB}$0are the cohesion free energy (Δ*G_sls_*), which gives an indication of the surface hydrophilicity and stability of a material immersed in a liquid medium.

#### 2.5.3. Interaction Energy

In this study, the PES UF membrane and SiO_2_ NP–NOM mixture were regarded as infinite planar surfaces and spherical colloids, respectively. According to the Derjaguin’s technique, the LW, AB, and EL interaction energy components between membrane and foulant can be written as follows:(9)$$U_{\mathrm{mlf}}^{\mathrm{LW}}{=2\pi d}_{0}^{2}{\Delta G}_{d_{0}}^{\mathrm{LW}}(\frac{a}{d})$$
(10)$$U_{\mathrm{mlf}}^{\mathrm{AB}}{=2\pi a\lambda \Delta G}_{d_{0}}^{\mathrm{AB}}\exp(\frac{d_{0}-d}{\lambda })\;$$
(11)$$U_{\mathrm{mlf}}^{\mathrm{EL}}{=\pi \varepsilon }_{0}\varepsilon_{r}a\left[{2\xi }_{m}\xi_{f}\ln(\frac{{1+e}^{-\kappa d}}{1-e^{-\kappa d}}{)+(\xi }_{m}^{2}{+\xi }_{f}^{2})\ln(1-e^{-2\kappa d})\right]$$
where *a* is the radius of the SiO_2_ NP–NOM mixture, *d* is the separation distance between the UF membrane and SiO_2_ NP–NOM mixture, *λ* (0.6 nm) is the characteristic decay length of the AB interaction, *ε*_0_ (8.854 × 10^−12^ C·V^−1^·m^−1^) is the permittivity of vacuum, *ε_r_* (78.5) is the relative permittivity of water, and *ξ_m_* and *ξ_f_* are the surface potentials of the UF membrane and SiO_2_ NP–NOM mixture, respectively. *κ* is the inverse Debye screening length, which can be obtained as follows:(12)$$\kappa =\sqrt{\;\;\frac{e^{2}\sum n_{i}z_{i}^{2}}{\varepsilon_{r}\varepsilon_{0}\mathrm{kT}}}$$
where *e* (1.6 × 10^−19^ C) is the electron charge, *n_i_* is the number concentration of ion *i* in the bulk solution, *z_i_* is the valence of ion *i*, *k* (1.381 × 10^−13^ J·K^−1^) is Boltzmann’s constant, and *T* is the absolute temperature. 

Similarly, the interfacial interaction energies between two spherical foulants are also determined using Derjaguin’s technique:(13)$$U_{\mathrm{flf}}^{\mathrm{LW}}{=2\pi d}_{0}^{2}{\Delta G}_{d_{0}}^{\mathrm{LW}}\frac{a_{1}a_{2}}{d(a_{1}+a_{2})}$$
(14)$$U_{\mathrm{flf}}^{\mathrm{AB}}=\frac{2\pi a_{1}a_{2}}{a_{1}+a_{2}}{\lambda \Delta G}_{d_{0}}^{\mathrm{AB}}\exp(\frac{d_{0}-d}{\lambda })$$
(15)$$U_{\mathrm{flf}}^{\mathrm{EL}}{=\pi \varepsilon }_{0}\varepsilon_{r}\frac{a_{1}a_{2}}{a_{1}+a_{2}}\xi_{f}^{2}{\ln(1+e}^{-\kappa d})$$
where *a_1_* and *a_2_* are the radii of the interacting foulant particles.

## 3. Results and Discussion

### 3.1. Effects of Solution Chemistry on UF Membrane Fouling by SiO_2_ NP–NOM Mixture

Figure 1 shows the PES UF membrane fouling caused by the SiO_2_ NP–NOM mixture, indicated by the decline of normalized flux as a function of permeate volume and induced under different solution chemistry conditions. It can be seen from Figure 1a that the extent of UF membrane fouling decreased with the increase of solution pH from 3 to 9. This can be attributed to the variation of surface charge and hydrophobicity of the SiO_2_ NP–NOM mixture and UF membrane at different solution pH levels. It is speculated that the negative charges of the SiO_2_ NP–NOM mixture and the UF membrane would be reduced at low pH levels, due to the protonation of carboxylic groups in the SiO_2_ NP–NOM mixture or sulfuryl groups in the PES membrane [27]. This would weaken the electrostatic repulsion between the SiO_2_ NP–NOM mixture and the UF membrane, and consequently make it easier for the mixture to attach to the membrane surface. In addition, owing to the reduced ionization of carboxylic and phenolic functional groups with decreasing pH, the SiO_2_ NP–NOM mixture would become more hydrophobic, enhancing the hydrophobic interaction between the SiO_2_ NP–NOM mixture and the UF membrane and thereby resulting in more serious membrane fouling.

No efforts have yet been devoted to understanding the effect of pH on SiO_2_ NP–NOM mixture fouling of UF membranes. Only a few studies have investigated the effect of pH on SiO_2_ NP–SA mixture fouling during FO or NF processes. For example, Kim et al. [29] and Xia et al. [34] conducted FO and NF of SiO_2_ NP–SA mixtures under various pH conditions, and reported that the extent of flux decline was minimal at pH = 7 and became more severe under either acidic or alkaline conditions. This is different from the variation trend displayed in Figure 1a and can be explained by the different properties of SA and NOM. When the solution pH is 7, the carboxylic groups of SA are fully deprotonated. In contrast, the degree of deprotonation of NOM (mainly HA) is enhanced with the increase of pH values from 3 to 9 due to the greater abundance of carboxylic and hydroxyl groups [35,36,37]. Therefore, the surface charge and hydrophilicity of NOM would increase continuously with the increase of solution pH. 

It can be seen in Figure 1b that PES UF membrane fouling by the SiO_2_ NP–NOM mixture was significantly aggravated with increasing ionic strength. For instance, in the 100 mM ionic strength experiment, the corresponding normalized flux decline at the end of filtration was 79.8%, which was much higher than the 49.7% obtained in the case of 10 mM ionic strength. More severe membrane fouling at elevated ionic strength has also been observed in NF of SiO_2_ NP–NOM mixtures [17]. At high ionic strengths, both SiO_2_ NP–NOM mixtures and UF membranes become less negatively charged due to double-layer compression and charge screening. This leads to the reduction of electrostatic repulsion between the UF membrane and the SiO_2_ NP–NOM mixture. Meanwhile, the hydrophobicity of the SiO_2_ NP–NOM mixture would increase in association with the reduced electrical charge, enhancing the hydrophobic interaction between the UF membrane and the SiO_2_ NP–NOM mixture. Consequently, a SiO_2_ NP–NOM mixture at high ionic strength would more easily attach to a UF membrane and result in more serious membrane fouling.

As shown in Figure 1c, the decline of normalized flux was quite responsive to the increase of calcium concentration. As expected, the addition of calcium ions caused a significant increase in normalized flux decline. Our results are consistent with those reported by Kim et al. [38] and Xia et al. [34] in their studies of the effect of calcium concentration on SiO_2_ NP–SA fouling during FO and NF, respectively. Unlike the case of ionic strength, however, the magnitude of the unfavorable effect of calcium concentration on UF membrane fouling by the SiO_2_ NP–NOM mixture appeared to be significantly larger. More specifically, the normalized flux of the UF membrane at 1.0 mM calcium concentration declined by 93.9% at the end of filtration, which was nearly 1.2 times higher than the decline in the case of 100 mM ionic strength (79.8%). It has been reported that calcium chloride can reduce the net charge of NOM significantly, not only in the same way as does the ionic strength of sodium chloride, but also due to the specific binding of calcium ions with the carboxylic groups of NOM [39,40,41]. It is speculated that the relevant interpretations in the literature may also be applicable to this study, due to the presence of NOM in the foulant mixture. The decrease in electrostatic repulsive interaction and increase in hydrophobic interaction, therefore, led to an increase in the attachment of the SiO_2_ NP–NOM mixture to the UF membrane, accompanied by more serious membrane fouling. 

To verify the speculations proposed above, the surface charge and hydrophobicity of the SiO_2_ NP–NOM mixture and UF membrane were calculated under different solution conditions as described in the following section. The major interfacial interactions (i.e., LW, AB, and EL interactions) involved in membrane fouling were subsequently analyzed using the xDLVO theory, based on which the underlying mechanisms governing PES UF membrane fouling by the SiO_2_ NP–NOM mixture were elucidated. 

### 3.2. Effects of Solution Chemistry on Physicochemical Properties of UF Membrane and SiO_2_ NP–NOM Mixture

The charge characteristics of the PES UF membrane and the SiO_2_ NP–NOM mixture were proportional to the electrostatic interactions of the membrane–foulant and foulant–foulant pairs [42,43,44]. Therefore, Figure 2 displays the changes in surface charge, indicated by zeta potential, of both the PES UF membrane and the SiO_2_ NP–NOM mixture as functions of solution pH, ionic strength, and calcium concentration. It is clear that both the UF membrane and the SiO_2_ NP–NOM mixture exhibited negatively charged surface characteristics under all solution conditions investigated. This indicates that electrostatic repulsive interactions would develop between the UF membrane and the SiO_2_ NP–NOM mixture, or among SiO_2_ NP–NOM molecules. As shown in Figure 2, the surface charge of the UF membrane was greatly influenced by solution chemistry conditions. Regarding pH values, at pH 3, the initial zeta potential value was only −24.7 mV, while this value dramatically decreased to −46.8 mV at pH 9. This decline at higher pH levels could be attributed to the increased adsorption of hydroxide ions (OH^−^) under alkaline conditions, and was also reported by Mo et al. [40]. With increasing ionic strength from 10 to 100 mM, there was a significant increase in the zeta potential of the UF membrane, which was mainly caused by the charge screening of sodium ions. The same relationship was also found by Motsa et al. [43]. When the calcium ion concentration increased from 0 to 1.0 mM, the surface zeta potential of the UF membrane increased from −38.2 to −18.7 mV. This change is reasonable because of the neutralization of negative surface charge by specific adsorption of calcium ions onto the UF membrane.

Although the impacts of solution chemistry on the surface charges of SiO_2_ NPs and NOM alone have been well documented in previous studies [30,38,45,46,47,48], information on SiO_2_ NP–NOM mixtures under different solution condition is still lacking. It can be seen in Figure 2a that, with increasing pH, the SiO_2_ NP–NOM mixture surface became strongly charged, as reflected in the higher absolute value of negative zeta potential. On the one hand, this can be mainly ascribed to the intensified deprotonation of the carboxylic and silanol groups on the surface of the SiO_2_ NP–NOM mixture at higher solution pH levels. On the other hand, higher pH values represent an increased OH^−^ concentration, along with an enhanced adsorption capacity of OH^−^ onto the SiO_2_ NP–NOM mixture. Moreover, compared with the reported data of zeta potentials of SiO_2_ NPs [45,48], the SiO_2_ NP–NOM mixture became more negatively charged, confirming the existence of surface interactions between the NOM and the dispersed inorganic nanoparticles. This is consistent with the results from Taheri [49] and Schulz [15], wherein SiO_2_ NPs had a higher negative surface charge in the presence of NOM. As the ionic strength increased, the zeta potential of the SiO_2_ NP–NOM mixture became less negative. In fact, it is generally acknowledged that the increasing zeta potential of NOM is mainly attributed to the effects of electrostatic double-layer compression and the shielding effect due to the increase in counter-ions [50]. As shown in Figure 2c, the absolute value of zeta potential gradually decreased with the increase in calcium ion concentration, implying that the SiO_2_ NP–NOM mixture will become more unsteady and more accessible for aggregation at higher concentrations. This phenomenon is in agreement with previous studies, which reported that calcium ions could reduce the negative charge of SiO_2_ NPs or NOM [27,30,46]. This is attributed partly to double-layer compression and charge screening, and partly to the complexation of calcium ions with the carboxylic groups on the surface of the SiO_2_ NP–NOM mixture resulting in charge neutralization. The profiles of zeta potential change with solution chemistry in Figure 2 are quite consistent with those of normalized flux in Figure 1, indicating that surface charge may play key roles in the effects of solution chemistry on PES UF membrane fouling by SiO_2_ NP–NOM mixtures. However, there are other factors that further influence the interactions between UF membranes and SiO_2_ NP–NOM mixtures and among SiO_2_ NP–NOM molecules, as is discussed in the following paragraphs.

### 3.3. Effects of Solution Chemistry on Surface Tension Parameters and Free Energy of Cohesion of UF Membrane and SiO_2_ NP–NOM Mixture

It is well known that the hydrophilic/hydrophobic characteristics of membranes and foulants have an important effect on membrane fouling [42,44,51]. Therefore, the contact angles of the PES UF membrane and the SiO_2_ NP–NOM mixture at different solution chemistries were measured and the results are summarized in Table 1. It can be seen from Table 1 that increasing pH reduced the ultrapure water contact angles (*θ_W_*) and glycerol contact angles (*θ_G_*) of the PES UF membrane, suggesting that the chemistry of the PES UF membrane was significantly altered, apparently becoming more polar and hydrated. This may be attributed to the formation of more intensive hydrogen bonds between water molecules and the PES UF membrane with the increase of OH^−^ at higher pH levels. Moreover, the deprotonation of the membrane surface is accelerated by the increased pH, promoting the capacity to supply electrons, which is also a mechanism for the effect of the pH value on the contact angle [52]. Our results are consistent with those reported by Meng [53], who measured the *θ_W_* of UF membranes at different pH levels. In addition, the *θ_W_* and *θ_G_* of the PES UF membrane increased with increasing in ionic strength and calcium concentration, indicating that its wettability with polar liquids was weakened. In contrast, no regular trend was observed for the diiodomethane contact angles (*θ_D_*) of the PES UF membrane. As shown in Table 1, the contact angles (*θ_W_*, *θ_G_*, and *θ_D_*) of the SiO_2_ NP–NOM mixture exhibited similar trends with solution chemistry to those of the PES UF membrane.

Table 2 presents the calculated surface tension parameters and free energy of cohesion (Δ*G_sls_*) of the PES UF membrane and the SiO_2_ NP–NOM mixture under different solution conditions. There was no significant difference in the Lifshitz–van der Waals components (*γ^LW^*) under the varied solution conditions for either the UF membrane or the SiO_2_ NP–NOM mixture, which could be attributed to the rarely changed apolar liquid contact angle, the only changeable parameter that *γ^LW^* relies on. The UF membrane had high electron-donor components and low electron-acceptor components under the solution conditions studied, corresponding to its relatively lower AB components. This result agrees with the results of previous studies [21,32] which have found that polymeric membranes are typically characterized by a high electron-donor monopolarity. Similarly to the UF membrane, the SiO_2_ NP–NOM mixture was characterized as having high electron-donor monopolarity, which is consistent with the results of previous studies focusing on NF filtration [27,54]. Moreover, the electron-donor components of the UF membrane and the SiO_2_ NP–NOM mixture increased along with pH, but lowered at higher ionic strengths and calcium concentrations. The former may be due to the deprotonation of surface groups at higher pH levels and the “saponification” behavior caused by the interactions of the alkaline solution and polymer interface, while the latter could be attributed to the enhanced electrostatic shielding function. 

Δ*G_sls_* represents the interaction free energy (per unit area) when two surfaces of the same material are immersed in a solvent (in this case, water) and brought into contact [21,31,55]. This value provides a quantitative insight regarding the hydrophobicity/hydrophilicity of the membrane and foulant. It can be seen from Table 2 that the Δ*G_sls_* value of the UF membrane gradually increased with increasing pH or decreasing ionic strength and calcium concentration, indicating the enhanced hydrophilicity of the UF membrane. This is in accordance with the trend of the ultrapure water contact angle shown in Table 1. For a hydrophobic membrane surface, more hydrophobic or less hydrophilic foulants would more easily attach to the membrane, resulting in a stronger flux decline. The negative Δ*G_sls_* value of the SiO_2_ NP–NOM mixture significantly decreased with the increase of pH, implying that the attachment of the SiO_2_ NP–NOM mixture to the membrane fouled by SiO_2_ NP–NOM mixture would be weakened with increasing pH. In contrast, the increase of ionic strength and the addition of calcium ions increased the negative Δ*G_sls_* value of the SiO_2_ NP–NOM mixture, suggesting an enhanced attachment of the SiO_2_ NP–NOM mixture to the fouled membrane. It is speculated that the high ionic strength and calcium concentration would induce a small screening length and a strong screening of surface charge, making the SiO_2_ NP–NOM mixture less soluble in water and more attractive to itself. Moreover, it was found that Δ*G_sls_* exhibited the same variation trend as the electron-donor components with solution chemistry, indicating that the hydrophilicity/hydrophobicity of both the PES UF membrane and the SiO_2_ NP–NOM mixture may be determined by electron-donor components.

### 3.4. Interaction Energies between UF Membrane and SiO_2_ NP–NOM Mixture

According to the cake filtration model, the entire filtration period can be divided into two periods (i.e., initial stage and later stage, corresponding to adhesion and cohesion stages, respectively). In the later stage, when the entire membrane surface is covered with foulant, the membrane fouling behavior is controlled by foulant–foulant interactions. To understand the effects of solution chemistry conditions on PES UF membrane fouling by the SiO_2_ NP–NOM mixture foulant, based on the surface tension parameters displayed in Table 2, Table 3 shows the LW, AB, EL, and total interfacial interaction energies of both the membrane–foulant and foulant–foulant pairs. According to the xDLVO theory, a negative value of interaction energy represents an attractive interaction that aggravates membrane fouling, while a positive value indicates a repulsive interaction hindering the same [24,56]. In addition, higher absolute values of positive/negative interfacial interaction energy signify a stronger repulsive/attractive interaction force between two surfaces immersed in liquid.

As shown in Table 3, both membrane–foulant and foulant–foulant pairs had negative LW and AB interaction energies regardless of the solution chemistry conditions, whereas the EL interaction energy remained positive in all studied circumstances. Thus, the LW and AB components persistently provided attractive forces accelerating membrane fouling under all studied solution conditions, and the EL component a repulsive force decelerating membrane fouling. The absolute value of the AB interaction energy, however, was much higher compared with the LW and EL interaction energies, resulting in the AB interaction energy playing the most important role in determining both the signs and absolute values of the overall interaction energies. Similar relationships among the interaction energies were also reported by Tao et al. [24] who calculated the interaction energy per unit between a PVC UF membrane and HA–FA mixture foulants at different pH values. Hence the characteristic of total interaction energy was mainly dominated by AB components, leading to consistently negative values of total interaction energies observed in all studied cases regardless of filtration periods or solution concentrations.

By comparing the later cohesion stage with the initial adhesion stage of membrane fouling, the decreased absolute value of the negative LW, AB, and total interaction energies indicated that attractive LW, AB, and total interaction forces would be weakened when the membrane surface was covered by foulant. The weakened LW interaction energy could be attributed to the higher *γ^LW^* of the SiO_2_ NP–NOM mixture, while the increased AB interaction energy might have been caused by the stronger deprotonation of the SiO_2_ NP–NOM mixture compared with the PES membrane. Moreover, the positive EL interaction energy significantly decreased by more than 86.64%. As the zeta potential of the SiO_2_ NP–NOM foulant was higher than that of the PES membrane under almost all studied solution conditions (Figure 2), the electrostatic repulsive force between foulant and foulant should be inferior to the force of the membrane–foulant combination, which could be responsible for this sharp decline. 

According to Table 3, the total interfacial interaction energy of the later stage in response to solution conditions shared a similar trend with the initial stage, and so did the LW, AB, and EL components. However, with the increase of pH, ionic strength, or calcium ion concentration, the absolute values of LW, AB, EL, and total interaction energy showed varied tendencies. More particularly, positive EL interaction energy increased when pH increased from 3 to 9, while the absolute values of negative LW and AB interaction energy decreased. The absolute value of negative total interaction energy also decreased, which indicated a weakened attractive force, which was in good accordance with the decelerated membrane fouling at higher pH shown in Figure 1. The weakened attractive total interaction energy with increasing pH between membrane and foulant was in good accordance with the literature [24,57]. When ionic strength increased from 10 to 100 mM, AB, EL, and total interaction energy showed an opposite trend compared with the case of pH: the absolute values of negative AB and total interaction energy increased, and the positive EL interaction energy decreased. In contrast, the absolute value of negative LW interaction energy was relatively lower under higher ionic strength conditions, as was the case for pH. When calcium ion concentration increased from 0 to 1.0 mM, AB, EL and total interaction energy showed a similar trend with ionic strength. Similarly, the absolute values of negative AB and total interaction energy increased with increasing calcium ion concentration, and the positive EL interaction energy decreased. The same trend for AB, EL, and total interaction energy was also found by Tao et al. [24], who studied the interaction mechanisms of HA membrane fouling at different calcium ion concentrations. The absolute value of the negative LW component, however, was increased at higher calcium concentrations, which was different from the cases of pH and ionic strength. The lower total interface interaction energy at higher ionic strengths or calcium concentrations was also consistent with the accelerated membrane fouling.

To further elucidate the roles of both solution conditions and separation distance in membrane fouling, the variation in interaction energies of membrane–foulant and foulant–foulant combinations with separation distance under different solution conditions are displayed in Figure 3. According to the xDLVO theory, the existence of an energy barrier means that foulants must have sufficient kinetic energy to overcome the barrier to reach the surface of a clean or a fouling membrane [58]. It can be seen from Figure 3 that significant energy barriers existed at separation distances shorter than 3 nm at the case of the membrane–foulant combination, while they disappeared for the foulant–foulant pairs, indicating that foulant was much easier to attach to the membrane in the later stage of filtration than the initial stage. 

Figure 3a shows that the energy barrier between membrane and foulant existed under both alkaline and acid conditions, though it decreased with the reduction of pH from 10 to 2. The same trend under alkaline conditions was also found in a previous study of the combination of PES MF membranes and HA–BSA mixtures, but the disappearance of the energy barrier under acid conditions was also reported [27]. The existence of an energy barrier under acid conditions in this study could be attributed to the stronger deprotonation of the SiO_2_ NP–NOM mixture, greatly influencing the interaction of the AB component. It can be seen in Figure 3c that the energy barrier decreased with the increase in ionic strength from 10 to 30 mM, and then disappeared with the further increase in ionic strength to 60 and 100 mM. Similarly, in Figure 3e, although the energy barrier almost disappeared at the calcium ion concentration of 1.0 mM, a lower value at higher calcium concentrations was also found. Thus, according to the above trends of total interaction energies with solution conditions, it can be concluded that the SiO_2_ NP–NOM mixture was subject to greater repulsive interactions with the PES UF membrane at higher pH levels, and lower ionic strengths and calcium ion concentrations, which fits well with the membrane fouling potential depicted in Figure 1.

### 3.5. Correlation between Fouling Potential and Interaction Energy

The entire membrane fouling process caused by the SiO_2_ NP–NOM mixture was separated into two stages (i.e., initial fouling stage and later fouling stage), based on the cake filtration model [49,59]. It is generally accepted that the initial stage of membrane fouling is dictated by the membrane–foulant interaction energy [60,61]. Over time, the membrane surface eventually became covered with SiO_2_ NP–NOM mixture, and, consequently, the interaction energy among SiO_2_ NP–NOM molecules governed the later membrane fouling. In Figure 4, the fouling potential of the initial and later fouling stages at each given solution chemistry is plotted against the corresponding interaction energies of membrane–foulant and foulant–foulant combinations. In this study, the initial fouling potential and later fouling potential were calculated from the decrease in normalized flux caused by the accumulated permeate volume at the initial and later fouling stages, respectively. 

It can be seen from Figure 4 that there was a clear negative linear correlation between fouling potential and interaction energy, regardless of the fouling stage and solution chemistry. In other words, the more negative the value of interaction energy, the higher the fouling potential. The results observed here are in good agreement with those of previous studies, wherein the extent of fouling was also plotted as a function of interaction energy [23,27,47]. The linear fitting to experimental data shows that the correlation coefficients were all above 0.7, further validating the critical role of membrane–foulant and foulant–foulant interactions in the initial and later membrane fouling stages, respectively. In other words, the xDLVO theory can be used to quantitatively elucidate the mechanisms of the effect of solution chemistry on UF membrane fouling caused by a SiO_2_ NP–NOM mixture. Furthermore, the correlation coefficients at the initial fouling stages were higher than those at later fouling stages in the examined pH and ionic strength ranges, implying that the xDLVO theory is more applicable to the interpretation of initial membrane fouling. The case of calcium concentration, however, was opposite to those of pH and ionic strength.

The slope of the linear trend line provides an indication of the change in the fouling potential for given variations in interaction energy. As shown in Figure 4, the initial fouling stage exhibited greater variations in membrane fouling behavior associated with changes in unit interaction energy as compared to the later fouling stage. This indicates that the membrane fouling behavior caused by the SiO_2_ NP–NOM mixture was more sensitive to the interaction energy in the initial fouling stage of the UF process.

## 4. Conclusions

In this study, the PES UF membrane fouling caused by a SiO_2_ NP–NOM mixture was investigated under different solution conditions, i.e., pH value, ionic strength, and calcium concentration. The corresponding fouling mechanisms were elucidated using the xDLVO theory, and the correlations between fouling potential and interaction energy were analyzed. The key findings are summarized as follows.

(1)The PES UF membrane fouling by the SiO_2_ NP–NOM mixture increased with decreasing pH, increasing ionic strength, and increasing calcium concentration, and the most severe membrane fouling was obtained at pH = 3.0, ionic strength = 100 mM, and calcium concentration = 1.0 mM. (2)Variations of the zeta potentials of both membrane and foulants with solution conditions showed the same trends, which was due to the similar deprotonation effect of functional groups and the electrostatic shielding effect.(3)Both the PES UF membrane and the SiO_2_ NPs–NOM mixture exhibited stronger hydrophobicity at lower pH, higher ionic strength, and higher calcium concentration.(4)At the minimum equilibrium cut-off distance (0.158 nm), the attractive AB interaction (negative value of AB interaction energy) between clean/fouled membrane and foulant was the major fouling mechanism in both the initial adhesion and later cohesion stages, while the attractive LW and repulsive EL interactions were of less importance to the total interaction energy.(5)The change of fouling potential with solution chemistry was found to be negatively correlated with the calculated total interaction energy, indicating that the PES UF membrane fouling behavior by the SiO_2_ NP–NOM mixture under different solution conditions can be effectively explained and predicted using the xDLVO theory.

## Figures and Tables

**Figure 1 membranes-13-00449-f001:**
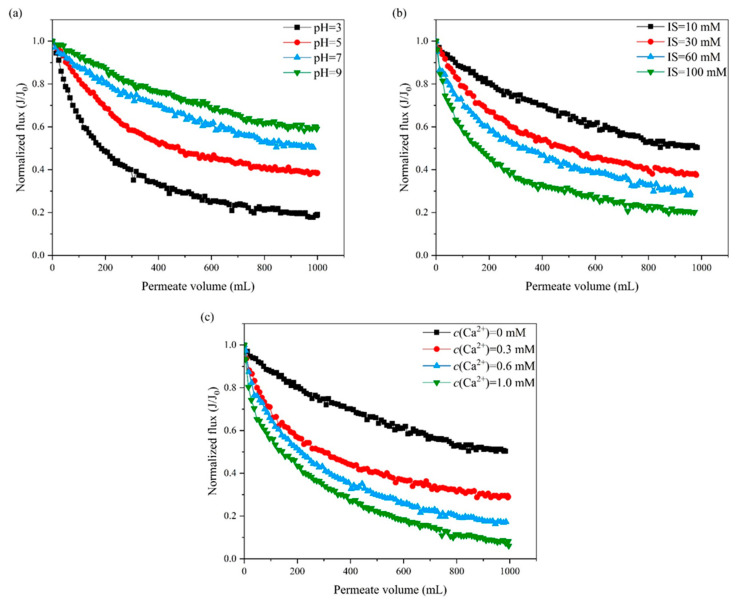
Variation of normalized flux with permeate volume during UF of SiO_2_ NP–NOM mixture under different solution conditions: (**a**) pH, (**b**) ionic strength, and (**c**) calcium concentration.

**Figure 2 membranes-13-00449-f002:**
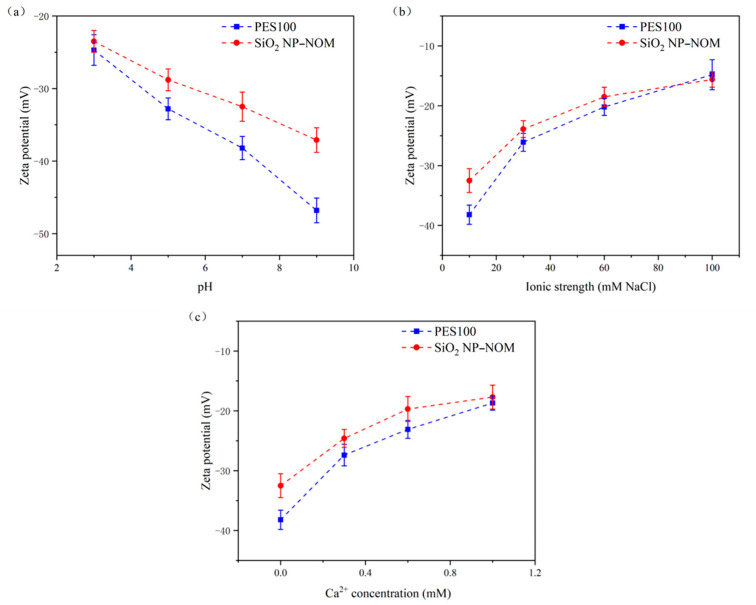
Variation of zeta potential of UF membrane and SiO_2_ NP–NOM mixture as a function of solution chemistry: (**a**) pH, (**b**) ionic strength, and (**c**) calcium concentration.

**Figure 3 membranes-13-00449-f003:**
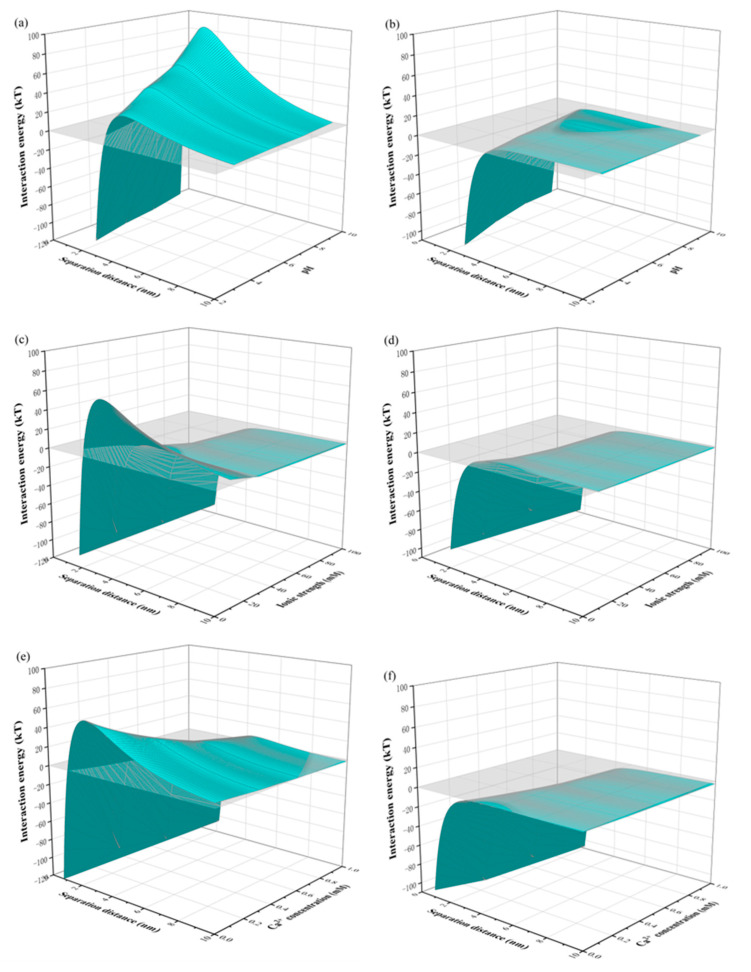
Profiles of the total interaction energies of membrane–foulant and foulant–foulant combinations with separation distance under different solution conditions: (**a**,**b**) pH, (**c**,**d**) IS, and (**e**,**f**) Ca^2+^ concentration.

**Figure 4 membranes-13-00449-f004:**
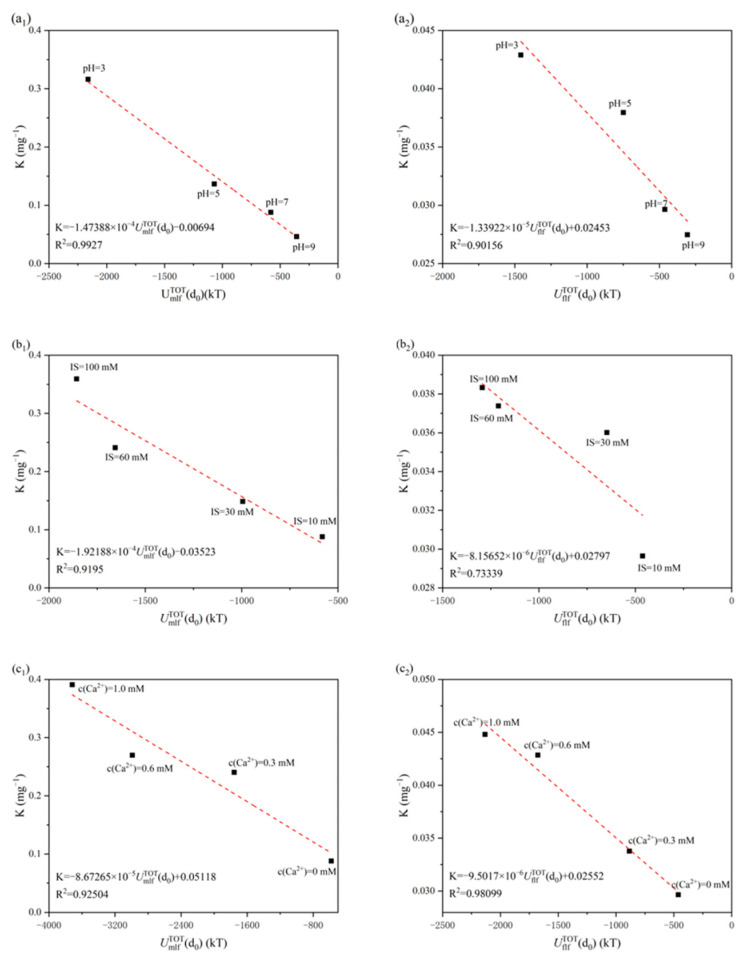
Correlation between fouling potential and interaction energy under different solution conditions: (**a_1_**,**a_2_**) pH, (**b_1_**,**b_2_**) ionic strength, and (**c_1_**,**c_2_**) calcium concentration. Subscript 1 represents initial fouling (adhesion) stage, subscript 2 represents later fouling (cohesion) stage.

**Table 1 membranes-13-00449-t001:** Measured contact angles of UF membrane and SiO_2_ NP–NOM mixture at different solution chemistries.

Solution Chemistry	PES UF Membrane			SiO_2_ NP–NOM		
	*θ_W_*(°)	*θ_G_*(°)	*θ_D_*(°)	*θ_W_*(°)	*θ_G_*(°)	*θ_D_*(°)
pH = 3	64.5 ± 2.1	67.1 ± 2.1	21.5 ± 1.3	69.8 ± 2.5	67.6 ± 2.6	13.8 ± 0.9
pH = 5	59.5 ± 1.3	61.8 ± 2.2	23.5 ± 1.8	64.4 ± 1.0	63.7 ± 1.8	11.7 ± 1.1
pH = 7	54.3 ± 1.4	55.4 ± 2.0	22.9 ± 2.2	59.5 ± 1.2	58.6 ± 1.5	10.9 ± 0.9
pH = 9	53.5 ± 2.6	54.3 ± 1.4	23.9 ± 1.6	53.1 ± 2.0	49.8 ± 1.4	11.5 ± 1.0
IS = 10 mM	54.3 ± 1.4	55.4 ± 2.0	22.9 ± 2.2	59.5 ± 1.2	58.6 ± 1.5	10.9 ± 0.9
IS = 30 mM	57.3 ± 1.8	56.5 ± 1.9	23.7 ± 1.6	62.6 ± 2.5	60.1 ± 1.5	12.5 ± 1.6
IS = 60 mM	61.1 ± 1.2	62.2 ± 1.1	20.6 ± 1.9	73.9 ± 1.5	68.5 ± 2.6	17.9 ± 2.0
IS = 100 mM	64.8 ± 1.8	65.8 ± 1.7	23.8 ± 2.2	78.3 ± 2.1	72.7 ± 1.8	16.4 ± 1.7
c(Ca^2+^) = 0 mM	54.3 ± 1.4	55.4 ± 2.0	22.9 ± 2.2	59.5 ± 1.2	58.6 ± 1.5	10.9 ± 0.9
c(Ca^2+^) = 0.3 mM	57.9 ± 1.5	60.1 ± 2.1	22.1 ± 1.9	63.4 ± 2.1	63.3 ± 1.7	14.4 ± 1.1
c(Ca^2+^) = 0.6 mM	65.4 ± 1.3	64.3 ± 1.1	25.2 ± 2.1	70.8 ± 2.4	69.2 ± 1.9	18.1 ± 1.7
c(Ca^2+^) = 1.0 mM	67.1 ± 1.7	66.3 ± 1.7	24.8 ± 1.9	76.8 ± 2.0	75.3 ± 1.9	16.4 ± 1.8

Notes: *θ_W_*: ultrapure water contact angle, *θ_G_*: glycerol contact angle, *θ_D_*: diiodomethane contact angle.

**Table 2 membranes-13-00449-t002:** Surface tension parameters (mJ/m^2^) and free energy of cohesion (mJ/m^2^) of UF membrane and SiO_2_ NP–NOM mixture under different solution conditions.

Solution Chemistry	PES UF Membrane						SiO_2_ NP–NOM					
	*γ^LW^*	*γ^−^*	*γ^+^*	*γ^AB^*	*γ^TOT^*	Δ*G_sls_*	*γ^LW^*	*γ^−^*	*γ^+^*	*γ^AB^*	*γ^TOT^*	Δ*G_sls_*
pH = 3	40.53	19.11	0.18	3.73	44.26	−18.31	49.34	14.09	0.30	4.14	53.48	−34.43
pH = 5	40.02	21.27	0.03	1.48	41.50	−14.03	49.75	17.98	0.19	3.72	53.47	−26.31
pH = 7	40.18	22.80	0.02	1.22	41.40	−10.97	49.89	20.50	0.04	1.73	51.62	−21.59
pH = 9	39.92	23.01	0.04	1.86	41.78	−10.35	49.79	22.32	0.08	2.72	52.51	−17.60
IS = 10 mM	40.18	22.80	0.02	1.22	41.40	−10.97	49.89	20.50	0.04	1.73	51.62	−21.59
IS = 30 mM	39.97	20.24	0.02	1.40	41.37	−16.28	49.60	17.64	0.03	1.56	51.16	−27.84
IS = 60 mM	40.75	19.62	0.03	1.58	42.33	−17.98	48.37	10.17	0.17	2.62	50.99	−44.97
IS = 100 mM	39.94	17.99	0.08	2.40	42.34	−20.86	48.75	8.13	0.41	3.64	52.39	−49.48
c(Ca^2+^) = 0 mM	40.18	22.80	0.02	1.22	41.40	−10.97	49.89	20.50	0.04	1.73	51.62	−21.59
c(Ca^2+^) = 0.3 mM	40.38	21.87	0.01	1.00	41.38	−21.57	49.21	19.05	0.17	3.63	52.84	−23.72
c(Ca^2+^) = 0.6 mM	39.56	16.44	0.02	1.02	40.58	−24.79	48.32	14.12	0.37	4.55	52.87	−33.37
c(Ca^2+^) = 1.0 mM	39.67	15.86	0.05	1.81	41.48	−25.91	48.75	11.22	0.91	6.38	55.13	−38.55

**Table 3 membranes-13-00449-t003:** Interfacial interaction energies (kT) of membrane–foulant and foulant–foulant pairs at d_0_ under different solution conditions.

Solution Chemistry	Membrane–Foulant				Foulant–Foulant			
	$U_{\mathrm{mlf}}^{\mathrm{LW}}\left(d_{0}\right)$	$U_{\mathrm{mlf}}^{\mathrm{AB}}\left(d_{0}\right)$	$U_{\mathrm{mlf}}^{\mathrm{EL}}\left(d_{0}\right)$	$U_{\mathrm{mlf}}^{\mathrm{TOT}}\left(d_{0}\right)$	$U_{\mathrm{flf}}^{\mathrm{LW}}\left(d_{0}\right)$	$U_{\mathrm{flf}}^{\mathrm{AB}}\left(d_{0}\right)$	$U_{\mathrm{flf}}^{\mathrm{EL}}\left(d_{0}\right)$	$U_{\mathrm{flf}}^{\mathrm{TOT}}\left(d_{0}\right)$
pH = 3	−235.723	−2026.298	100.940	−2161.082	−163.542	−1306.611	12.182	−1457.971
pH = 5	−177.211	−1016.404	123.374	−1070.242	−127.486	−636.100	13.916	−749.669
pH = 7	−152.850	−565.573	137.097	−581.326	−109.582	−367.599	15.107	−462.075
pH = 9	−146.663	−393.764	182.169	−358.258	−106.142	−219.402	19.182	−306.362
IS = 10 mM	−152.850	−565.573	137.097	−581.326	−109.582	−367.599	15.107	−462.075
IS = 30 mM	−138.491	−919.045	62.940	−994.596	−99.423	−555.363	7.335	−647.451
IS = 60 mM	−134.475	−1557.187	35.696	−1655.966	−89.637	−1124.330	4.157	−1209.810
IS = 100 mM	−125.403	−1751.537	20.601	−1856.339	−87.854	−1208.955	2.752	−1294.057
c(Ca^2+^) = 0 mM	−152.852	−565.573	137.097	−581.326	−109.582	−367.599	15.107	−462.075
c(Ca^2+^) = 0.3 mM	−239.561	−1636.386	119.298	−1756.650	−166.711	−730.978	13.693	−883.995
c(Ca^2+^) = 0.6 mM	−255.633	−2824.642	90.672	−2989.602	−179.987	−1506.191	10.002	−1676.176
c(Ca^2+^) = 1.0 mM	−277.113	−3511.627	71.365	−3717.375	−196.689	−1944.243	8.571	−2132.360

Notes: LW, AB, and EL represent the Lifshitz–van der Waals, acid–base, and electrostatic contributions to energy, respectively, while *m*, *l*, and *f* represent the contacts involving membrane, liquid, and foulant, respectively; *d_0_* is the minimum equilibrium cut-off distance (0.158 ± 0.009 nm).

## Data Availability

Not applicable.

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
