# Peer review of "Quantitative Assessment of Interfacial Interactions Governing Ultrafiltration Membrane Fouling by the Mixture of Silica Nanoparticles (SiO2 NPs) and Natural Organic Matter (NOM): Effects of Solution Chemistry"

_membranes, 2023, doi:10.3390/membranes13040449_

Round 1

Reviewer 1 Report

Kindly refer to the attached file

Author Response

Response to Reviewer 1’s comments

Point 1: This paper studies the fouling of UF membrane when applying silica nano-particles with natural organic matters under different conditions of pH, ionic charge and concentrations.

Response 1: Your comment summarized the key points of our experiments. Thank you very much for your time to review this manuscript. 

Point 2: Abstract: well written, concise and provide complete information on the research findings with general details about the content of the paper.

Response 2: Thank you very much for your encouraging comments.

Point 3: In the keywords I suggest removing the solution chemistry from the list to avoid

ambiguity.

Response 3: Thank you for your suggestion. The keyword "solution chemistry" was deleted to avoid ambiguity (Page 1, Line 32).

Point 4: Introduction: well written and covers all the literature about the research however I think adding more information about the kinetics of the removal of SiO2 NPs-NOM would be of interest to the reader.

Response 4: Thank you for your suggestion. Since the kinetics of pollutant removal is not the concern of this work and, more importantly, there are no results to support it, we did not add the relevant information after careful consideration. Your kind understanding would be highly appreciated.

Point 5: The research question at the end of this part clarifies the objectives of the paper.

Response 5: Thank you very much for your encouraging comments.

Point 6: Materials and methods part: In line 120 check the concentration of HA and BSA (8.0, 2.0 and 2.0 mg).   

Response 6: Double-checked.

Point 7: I think adding a schematic diagram of the used unit would provide more insight on your experimental setup.

Response 7: Thank you for your suggestion. Since the dead-end stirred cell filtration system is so widely reported in the literature, it seems not necessary to show its schematic diagram again. Particularly, in our case, the schematic diagram is exactly the same as the one shown in our previous work published in Water, an open access journal.

Point 8: The Method is clear

Response 8: Thank you very much for your encouraging comments.

Point 9: The results are presented in an understandable logical progression and the discussion is supported with good references to the reader, however I strongly recommend adding more up-to-date references to your list.

Response 9: Thank you very much for your encouraging comments. According to your suggestion, the following references published in recent five years were added in the revised manuscript.

[1] Bharti S., Kumar S.M. Synergetic effects of organic and inorganic additives on improvement in hydrophilicity and performance of PVDF antifouling ultrafiltration membrane for removal of natural organic material from water. Journal of Applied Polymer Science, 2021, 138, 50568.

[2] Esfahani A.R., Zhang Z.. Sip Y.Y.L., et al. Removal of heavy metals from water using electrospun polyelectrolyte complex fiber mats. Journal of Water Process Engineering. 2020, 37, 101438.

[3] Marie E, Jingshi W., Andrea M., et al. Mitigation of membrane fouling by nano/microplastics via surface chemistry control. Journal of Membrane Science. 2021, 633, 119379.

[4] Lu D.W., Jia B.H., Xu S., Wang P.P., et al. Role of pre-coagulation in ultralow pressure membrane system for Microcystis aeruginosa-laden water treatment: Membrane fouling potential and mechanism. Science of the Total Environment 2020, 710, 136340.

Point 10: Conclusion is precise and addresses the findings of the paper in general

Response 10: Thank you very much for your encouraging comments.

Point 11: General Comments: I suggest adding a list of abbreviation to the paper to ease follow the symbols

Response 11: According to your suggestion, a list of abbreviation was added in the revised manuscript (Page 16-17, Line 576).

Reviewer #1: Based on these point I have no reservation publishing this paper after considering the minor comments highlighted above

Thank you very much for your encouraging comments which are of great importance for the further improvement of our work's quality and are seriously addressed or clarified to our very best.

Reviewer 2 Report

In this work the effect of SiO2 NPs-NOM solution chemistry on polyethersulfone (PES) UF membrane fouling was investigated at different pH, ionic strengths and calcium concentrations. For the first time the xDLVO theory is adopted to elucidate the corresponding membrane fouling mechanisms by quantitatively evaluating the interfacial interactions (i.e., LW, AB and EL interaction) between SiO2 NPs-NOM mixture and UF membrane, and among SiO2 NPs-NOM mixtures.

The work has been done at a high level and deserves publication in the "Membranes" after minor revisions.

1. As this acticle is about membrane fouling,  it would not be superfluous to mention in the introduction the methods leading to decrease of membrane fouling.

2. L.116. "...HA, BSA and BSA..." apparently it should be SA

3. In 2.4 Analytical Methods authors described the method of contact angle measurement. However, using the sessile drop method for UF membranes was not suitable, because of the porous membrane surface. Could the authors provide measurements of contact angle in time?

4.What was the technique of contact angle measurements in the case of SiO2 NPs-NOM mixture?

5. I suggest auhors to change the order of results description: p. 3.2 and 3.3 better to move above 3.1 for better undrstanding. In this case, the logic of data presentation will be more clearer and authors can use their data (zeta potential, contact angles, surface tension parameters and free energy of cohesion) to explain transport properties of UF membranes.

Author Response

Response to Reviewer 2’s comments

Reviewer #2: In this work the effect of SiO2 NPs-NOM solution chemistry on polyethersulfone (PES) UF membrane fouling was investigated at different pH, ionic strengths and calcium concentrations. For the first time the xDLVO theory is adopted to elucidate the corresponding membrane fouling mechanisms by quantitatively evaluating the interfacial interactions (i.e., LW, AB and EL interaction) between SiO2 NPs-NOM mixture and UF membrane, and among SiO2 NPs-NOM mixtures.

The work has been done at a high level and deserves publication in the "Membranes" after minor revisions.

Thank you very much for your time to review this manuscript, and we are glad to receive your constructive comments and positive evaluation. Your comments are of great importance for the further improvement of our work's quality and are seriously addressed or clarified to our very best. Please refer to the following point-to-point responses.

Point 1: 1. As this acticle is about membrane fouling,  it would not be superfluous to mention in the introduction the methods leading to decrease of membrane fouling.

Response 1: Thanks for your constructive suggestion. The following sentences describing the methods to mitigate membrane fouling were added in the revised manuscript (Page 2, Line 56-60).

"Moreover, the methods for mitigating UF membrane fouling have also been put forward, such as the control of solution chemistry having important effects on fouling behavior [16]. Nevertheless, in the case of UF membrane fouling by SiO2 NPs-NOM mixture, the solution chemistry has not been investigated."

Point 2: L.116. "...HA, BSA and BSA..." apparently it should be SA.

Response 2: Corrected.

Point 3: In 2.4 Analytical Methods authors described the method of contact angle measurement. However, using the sessile drop method for UF membranes was not suitable, because of the porous membrane surface. Could the authors provide measurements of contact angle in time?

Response 3:  We acknowledge that the sessile drop method does have certain limitations for measuring UF membrane contact angles. Nevertheless, it is kind of acceptable due to its wide application in the literature. Some references were listed below.

[1] Sunjin Kim, Noeon Park, Sungyun Lee, et al. Membrane characterizations for mitigation of organic fouling during desalination and wastewater reclamation. Desalination, 2008, 238 (1), 70-77.

[2] Prihandana, G. S., Ito, H., et al. Polyethersulfone Membrane Coated With Nanoporous Parylene for Ultrafiltration. Journal of Microelectromechanical Systems, 2012, 21(6), 1288-1290.

[3] Bing Zhang, Heli Tang, Dongmei Huang, et al. Effect of pH on anionic polyacrylamide adhesion: New insights into membrane fouling based on XDLVO analysis. Journal of Molecular Liquids, 2020, 32, 114463.

In our study, we performed contact angle measurements with the "newTV" software produced by the manufacture of the JC2000C Contact Angle Meter, a contact angle analyzer with video microscope. The measurement of contact angle was performed at the time of two second after the liquid drop formed on the membrane. The following sentence was added in the revised manuscript to indicate the time when the measurement of contact angle was performed in the revised manuscript (Page 4, Line 160-162).

“The measurement of contact angle was performed at the time of two second after the liquid drop formed on the clean PES UF membranes or the PES UF membranes cov-ered with SiO2 NPs-NOM mixture.”

Point 4: What was the technique of contact angle measurements in the case of SiO2 NPs-NOM mixture?

Response 4: According to the methods reported in the literature, the contact angles of SiO2 NPs-NOM mixture deposited on the PES UF membrane were measured using sessile drop method similarly as those of UF membrane. To further clarify this point, the following sentences and references were added in the revised manuscript (Page 4, Line 157-159).

"The SiO2 NPs-NOM mixture was firstly deposited on the PES UF membrane using the dead-end stirred cell filtration system, and then subjected to the contact angle measurement [30]."

Point 5: I suggest auhors to change the order of results description: p. 3.2 and 3.3 better to move above 3.1 for better undrstanding. In this case, the logic of data presentation will be more clearer and authors can use their data (zeta potential, contact angles, surface tension parameters and free energy of cohesion) to explain transport properties of UF membranes.

Response 5: Thank you for your suggestion. The structure of the Results and discussion was not adjusted after carefully considering your suggestion. Since there is a strong progressive logic relationship among the sections of 3.2, 3.3 and 3.4, the section of 3.2 and 3.3 cannot move above section 3.1. Without section 3.4 (interfacial interaction energies), the UF membrane fouling by SiO2 NPs-NOM mixture at different solution conditions cannot be explained. In fact, the logic structure of this work is basically the same with many other studies reported in the literature. It is fairly acceptable in this sense. 
